# Archaea produce peptidoglycan hydrolases that kill bacteria

Romain Strock[1,2], Valerie W.C. Soo[1,2,3,4], Pauline Misson[1,2,5], Georgia Roumelioti[1], Pavel V. Shliaha[1], Antoine Hocher[1,2,6], Tobias Warnecke[1,2,5,7]*

1 Medical Research Council Laboratory of Medical Sciences, London, United Kingdom, 2 Institute of Clinical Sciences, Faculty of Medicine, Imperial College London, London, United Kingdom, 3 Department of Infectious Disease, Faculty of Medicine, Imperial College London, London, United Kingdom, 4 Centre for Bacterial Resistance Biology, Imperial College London, London, United Kingdom, 5 Department of Biochemistry, University of Oxford, Oxford, United Kingdom, 6 Department of Genetics, University of Cambridge, Cambridge, United Kingdom, 7 Trinity College, Oxford, United Kingdom

* tobias.warnecke@bioch.ox.ac.uk

## Abstract

The social life of archaea is poorly understood. In particular, even though competition and conflict are common themes in microbial communities, there is scant evidence documenting antagonistic interactions between archaea and their abundant pro-karyotic brethren: bacteria. Do archaea specifically target bacteria for destruction? If so, what molecular weaponry do they use? Here, we present an approach to infer antagonistic interactions between archaea and bacteria from genome sequence. We show that a large and diverse set of archaea encode peptidoglycan hydrolases, enzymes that recognize and cleave a structure—peptidoglycan—that is a ubiquitous component of bacterial cell walls but absent from archaea. We predict the bacterial targets of archaeal peptidoglycan hydrolases using a structural homology approach and demonstrate that the predicted target bacteria tend to inhabit a similar niche to the archaeal producer, indicative of ecologically relevant interactions. Using a het-erologous expression system, we demonstrate that two peptidoglycan hydrolases from the halophilic archaeaon *Halogranum salarium* B-1 kill the halophilic bacterium *Halalkalibacterium halodurans,* a predicted target, and do so in a manner consistent with peptidoglycan hydrolase activity. Our results suggest that, even though the tools and rules of engagement remain largely unknown, archaeal-bacterial conflicts are likely common, and we present a roadmap for the discovery of additional antago-nistic interactions between these two domains of life. Our work has implications for understanding mixed microbial communities that include archaea and suggests that archaea might represent a large untapped reservoir of novel antibacterials.

**Data availability statement:** All data are available as part of the Supplementary Material. Processed public data and scripts used for processing can be found at zenodo.org/records/15534318 and github.com/srom/archaea-vs-bacteria.

**Funding:** This work was funded by core funding from the Medical Research Council (https://www.ukri.org/councils/mrc/, grant number: MC-A658-5TY40) to TW. VWCS is supported by an MRC CDA (MR/X007421/1) and AH by a Wellcome Trust CDA (227755/Z/23/Z). The funders played no role in the study design, data collection and analysis, decision to publish, or preparation of the manuscript.

**Competing interests:** The authors have declared that no competing interests exist.

**Abbreviations:** HMMs, Hidden Markov Models; PGB, peptidoglycan-binding; PGHs, peptidoglycan hydrolases.

## Introduction

Archaea are best known as extremophiles, thriving in environments where few other organisms survive. Such environments—from hot springs to salt lakes to acid mine drainage—are often dominated or even exclusively inhabited by archaea [1–3]. Over the last 10–15 years, however, broader and more sensitive sampling has revealed that archaea are not limited to such extreme environments. On the contrary, they are found in diverse locations, including the digestive tracts of cattle and termites, where they drive methane production, and various soil and marine ecosystems where they play key roles in global carbon and nitrogen cycles [4]. In all of these settings, archaea are surrounded—and typically outnumbered—by bacteria [5].

How archaea interact—physically, ecologically, and molecularly—with bacterial members of their respective communities is poorly understood [4]. What we do know largely concerns commensal or symbiotic associations, where at least one member of the community benefits from the interaction and nobody is harmed. Notably, this includes syntrophic consortia of methane-oxidizing archaea and sulphate-reducing bacteria in marine environments, where archaea consume end products of bacterial metabolism (e.g., acetate, hydrogen, or formate) that would otherwise inhibit bacterial growth [6,7]. Syntrophic interactions have also been observed inside eukaryotic hosts, including the guts of mammals [8–10], and likely underpin some intimate physical associations observed between archaea and bacteria [11–13].

One might suspect, however, that things are not always harmonious. For most niches, conflict is probably more common than cooperation [14,15]. Yet evidence for antagonistic interactions between archaea and bacteria (where one species harms the other, be that through competition, predation, or incidentally) is very limited. We are aware of only a handful of reports detailing antagonistic interactions, all involving supernatant from archaea grown in pure culture, which was found to inhibit the growth of (often ecologically unrelated) indicator bacteria [16–19]. None of these studies determined the mechanism of inhibition. Nor did they answer the question whether antibacterial activity had evolved as a specific adaptation or whether bacteria were simply innocent by-standers, killed in the crossfire of mechanisms evolved to target other archaea (or perhaps eukaryotes).

Does the dearth of reported cases of archaeal-bacterial conflict imply that archaea generally shun confrontation and that niche partitioning is largely frictionless and amicable? To us, this seems unlikely; archaea, after all, are not shy to defend their niche against other *archaea*, using a variety of proteins or small molecules [19–24].

How might one determine whether archaea antagonistically—and specifically—target bacteria? We reasoned that one might do so by looking, in archaeal genomes, for proteins that destructively target cellular structures that are unique to bacteria. One prominent such structure is peptidoglycan, a continuous mesh that envelops the cytoplasmic membrane of bacterial cells, composed of polysaccharide chains that are covalently crosslinked by peptide moieties. Some methanogenic archaea produce analogous extracellular structures of cross-linked glycans and peptides, but this pseudo-peptidoglycan (pseudomurein) exhibits different composition and linkage patterns, making it chemically distinct from bacterial peptidoglycan [25]. We therefore

decided to survey archaeal genomes for the presence of enzymes that cleave bacterial peptidoglycan: peptidoglycan hydrolases (PGHs).

PGHs are ubiquitous in bacteria. They often exhibit modular architectures (Fig 1A), with one or more peptidoglycan binding domains, which confer target specificity, fused to a catalytic domain (i.e., an amidase, peptidase, or glycosidase), which cleaves specific bonds of the peptidoglycan molecule. Bacteria deploy PGHs to remodel their own cell wall during cell division (autolysins), but occasionally also as weapons [26]. In these instances, following secretion, the PGH no longer acts on the producer's own peptidoglycan but diffuses to its target—often a closely related strain [27,28]—where it cleaves peptidoglycan to compromise cell wall integrity. Examples of such weaponized PGHs include lysostaphin from *Staphylococcus simulans* bv. staphylolyticus (Fig 1A) and zoocin A from *Streptococcus equi* subsp. *zooepidemicus* 4881, which act on a number of other Staphylococcus and Streptococcus strains, respectively [27,29].

We reasoned that, since archaea do not make peptidoglycan, the presence of peptidoglycan-cleaving proteins in archaeal genomes might suggest that these archaea interact with bacteria in an antagonistic manner. Indeed, one case of an archaeon-encoded PGH with antibacterial activity has previously been reported: tracing horizontal gene transfers of antibacterial genes across the tree of life, Metcalf and colleagues discovered a GH25 muramidase (UniProt ID: B5ID12) in the genome of *Aciduliprofundum booneii* [30]. A purified version of this PGH was shown to kill a range of Gram-positive bacteria in the family Bacillaceae, from which the enzyme was likely acquired. Below, we show that the case of *A. booneii* B5ID12 is not a collector's item and that PGHs with bactericidal activity are found in the genomes of diverse archaea, with implications for our understanding of how archaea act within and shape polymicrobial ecosystems.

## Results

### Archaeal genomes harbor diverse peptidoglycan hydrolases

We assembled and surveyed a phylogenetically balanced database of prokaryotic genomes (3,706 archaea; 50,640 bacteria, S1 Table) for proteins with homology to known PGHs (see Methods). Some catalytic domains involved in peptidoglycan processing can, in principle, cleave peptide or sugar bonds whether or not these bonds are specifically part of peptidoglycan. To increase the likelihood that peptidoglycan is the native substrate, we therefore restricted analysis to PGHs with a modular architecture, containing at least one catalytic and one cell wall-binding domain. We note that such modular architectures are more common in Gram-positive bacteria, where peptidoglycan is not surrounded by an outer membrane and peptidoglycan-binding domains prevent unwanted diffusion away from the cell. Our results are therefore likely weighted towards PGHs that originate from Gram-positive bacteria.

As expected, we find recognizable modular PGHs in a large proportion of bacterial genomes ($N = 42,612$, 84%). Surprisingly, however, we also find PGH homologs in nearly 5% ($N = 175/3,706$) of archaeal genomes, including representatives from most archaeal phyla and spanning a broad range of PGH architectures (Fig 1C and S2 Table). PGH homologs are particularly prominent (and diverse) amongst Nanoarchaeota, which are thought to lead mostly symbiotic or parasitic lives [31], Thermoplasmatota, and Halobacteriota. Only two of the PGH homologs come from the two classes of archaea known to synthesize pseudomurein (Methanobacteria and Methanopyri), even though those clades are well represented in our database ($N = 117$ genomes), suggesting that the large majority of PGHs identified here do not act on pseudomurein of their own producer.

While not the rule, it is not uncommon for a single archaeal genome to encode more than one PGH (12%, $N = 21/175$, Fig 1D and S2 Table).

For archaeal PGHs to act on bacterial peptidoglycan, they need to be secreted. We find that 32% ($N = 67/210$) of archaeal PGH homologs encode a known signal peptide (65% Sec; 35% Tat; Fig 1E and S2 Table), consistent with deployment outside the cell. Interestingly, archaeal PGHs *without* a recognized signal peptide are somewhat enriched (Fisher test $P = 0.059$) in genomes that also encode a phage sheath 1 domain (PF04984), indicative of the presence of a contractile injection system [24], which can in principle deliver PGHs as lethal cargo.

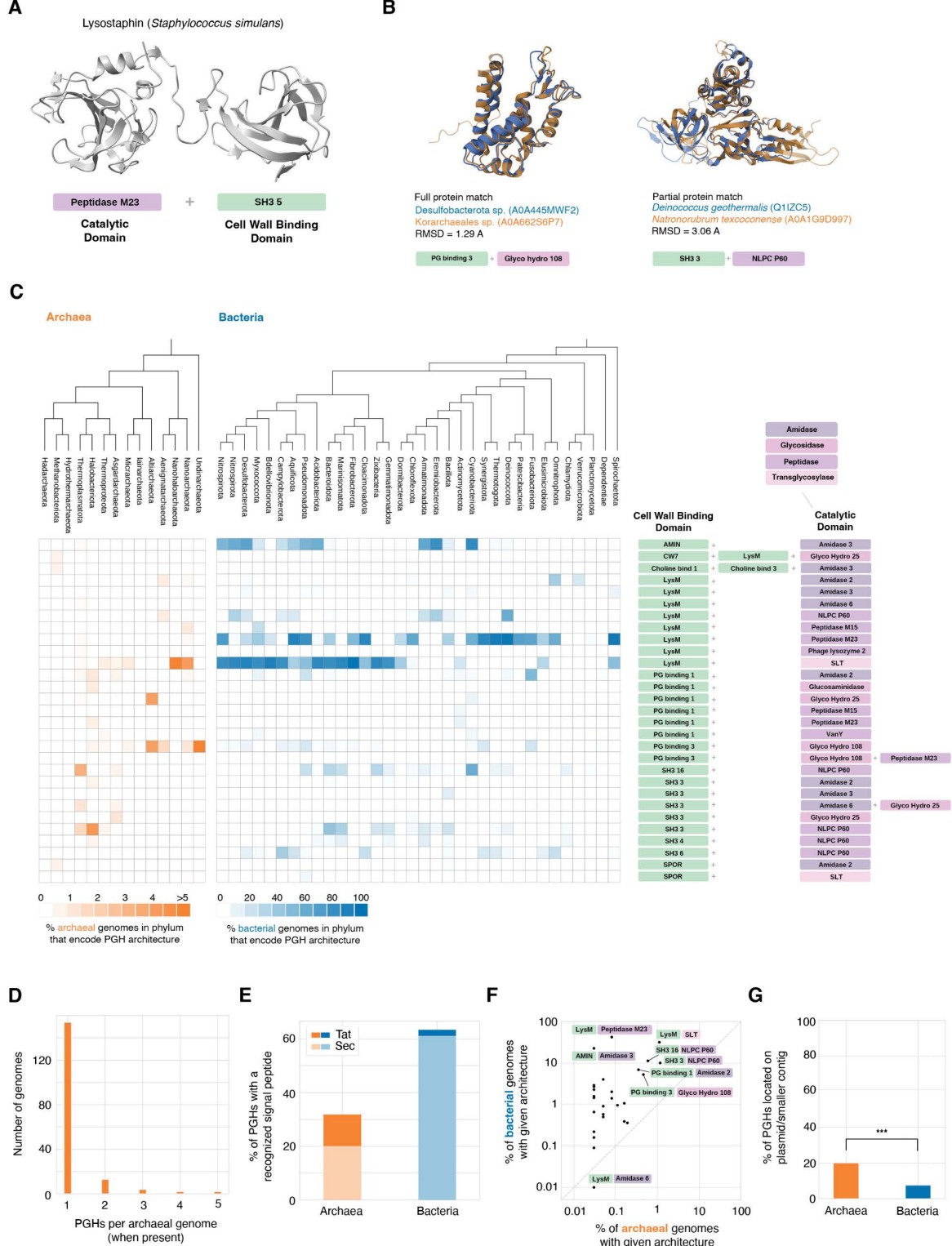

**Fig 1. Peptidoglycan hydrolases in archaea. A.** Structure of the peptidoglycan hydrolase (PGH) lysostaphin from *Staphylococcus simulans*, highlighting its modular domain architecture. **B.** Examples of structural homology between bacterial and archaeal PGH homologs, covering either the full (left) or parts of (right) the protein query. **C.** Distribution of modular PGH homologs across archaea and bacteria, summarizing the relative abundance of different

domain combinations at the phylum level. **D.** Number of archaeal genomes in the database that encode one or more modular PGH. **E.** Proportion of modular PGHs with a known signal peptide, split according to the predicted secretion system (Tat/Sec). **F.** Incidence of different modular PGH architectures in bacteria vs. archaea. **G.** Proportion of modular PGH homologs located on a plasmid (for completely assembled genomes) or, for incompletely assembled genomes, contigs below the median size of contigs in the assembly. \*\*\*$P = 8.9 * 10^{-7}$, Fisher's exact test. Code and data are available at zenodo.org/records/15534318 ("notebook/figure1.ipynb" and "data/figure1/", respectively).

In some instances, the binding domain of a PGH may interact with substrates other than peptidoglycan. For example, the LysM domain can, at least on occasion, bind chitin as well as peptidoglycan [32], and some archaea secrete glycosyl hydrolases to digest chitin extracellularly as an alternative carbon source [33]. In many cases, however, structural homology (see Fig 1B for examples), conservation of known catalytic residues, and the presence of both a peptidoglycan binding domain *and* a catalytic domain involved in peptidoglycan cleavage suggest peptidoglycan as the conserved substrate.

To investigate the functionality of archaeal PGH homologs in greater detail, we decided to focus on archaeal homologs of zoocin A, one of the best-described weaponized PGHs in bacteria [27,34–36]. We detect homology against the peptidase M23 domain of zoocin A in 10 archaeal genomes in our database of modular archaeal PGHs, including in five genomes belonging to the order Halobacteriales and four genomes from the class Nanoarchaeia (S2 Table). The cell wall binding domain of *S. equi* zoocin A (zoocin A target recognition domain, PF16775) is not found in archaea. Constructing a pan-prokaryotic phylogenetic tree of the M23 domains (see Methods), we find that the homologous domains found in Halobacteriales form two monophyletic groups, each branching with different bacteria from the phylum Bacillota (Fig 2A). In these archaea, the M23 domain lies N-terminal of two tandem peptidoglycan binding domains (PG binding 1 domain; PF01471). In the most closely related bacterial proteins, on the other hand, we find M23 on the C-terminal side of a LysM domain. These observations suggest an evolutionary history marked by domain shuffling, previously proposed as a hallmark of bacterial PGHs [37]. A phylogenetic tree built from the PG binding 1 domain supports this view, highlighting an entirely different set of bacteria, some of which exhibit the same domain composition found in archaea (Fig 2B).

The closest bacterial sister clade to the archaeal PG binding 1 domains contains a phylogenetic jumble of bacteria—including Actinomycetota, Chloroflexota, and Deinococcota—and a variety of domain architectures (Fig 2B) suggesting that horizontal transfer and domain shuffling have also been happening amongst bacteria. We tentatively suggest a single acquisition of a PG binding 1 domain-containing gene in the Halobacteriaceae, from a bacterial donor affiliated with the Actinomycetota, followed by repeated replacement of the catalytic domain, which generated the chimeric M23 domain-containing proteins that we focus on here.

Although we find PGHs with a M23 domain in several Halobacteriales genomes, the majority of Halobacteriales genomes in our database do not encode modular PGHs (S1 Fig). This patchy phylogenetic distribution, where homologs are present in some closely related genomes but absent from others, is not dissimilar from that of weaponized PGHs, and bacteriocins more generally, in bacteria [28], where it has been interpreted as consistent with evolution under an arms-race regime: PGH utility changes rapidly as targets adapt and new targets emerge, driving frequent gain and loss.

Transient utility of PGHs and their exchange via HGT is further supported by the fact that a significant fraction is found on plasmids versus main chromosomes (4/18 in archaea compared to 102/13,786 in bacteria; Fisher's exact test $p$-value < 1e−5; Fig 1G), or—for genomes that are incompletely assembled—enriched on smaller contigs, which are *a priori* more likely to correspond to plasmids or secondary chromosomes (26/133 in archaea versus 8,292/97,918 in bacteria; Fisher's exact test $p$-value < 1e−4; Fig 1G, see Methods).

Taken together, a patchy phylogenetic distribution (S1 Fig), biased localization on plasmids (Fig 1G), capacity for secretion (Fig 1E), and high levels of structural conservation (e.g., Fig 1B) are consistent with the idea that archaeal PGHs might be used in conflicts with bacteria.

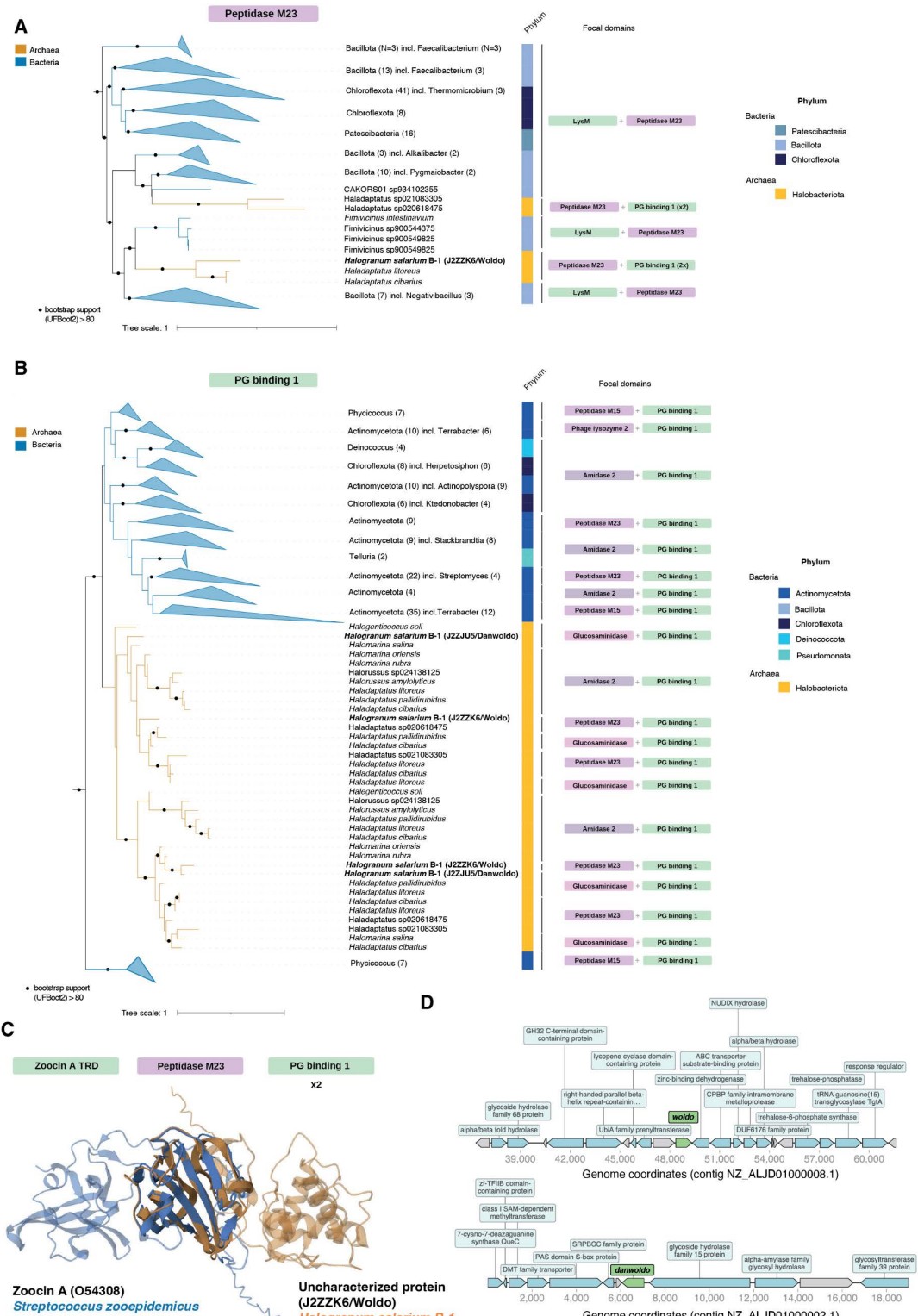

**Fig 2. Chimeric zoocin A homologs in archaea.** Protein domain-level phylogenetic trees highlighting evolutionary relationships of **(A)** peptidase M23 (Pfam ID: PF01551) and **(B)** PG binding 1 domains (Pfam ID: PF01471) from modular PGHs. The trees shown do not include all homologous domains present in archaea/bacteria but are subtrees that include all archaeal homologs from the Halobacteriota phylum and their immediate bacterial context.

Domains within each protein are treated individually; hence, proteins with two or more instances of the focal domain appear more than once in the tree. Tree scale: substitutions per site. See Methods for details on how trees were constructed. Note that domain composition describes the presence of a given domain but not its number. Most proteins here carry more than one copy of a particular cell wall binding domain, including Woldo, which has two tandem copies of the PG binding 1 domain. **C.** Structural overlay of zoocin A from *Streptococcus zooepidemicus* (UniProt ID: O54308) and Woldo, a candidate PGH from *Halogranum salarium* B-1 (UniProt ID: J2ZZK6), highlighting homologous M23 domains but divergent cell wall binding domains. TRD: target recognition domain. PG: peptidoglycan. **D.** Genomic context of *woldo* and *danwoldo* in the genome of *H. salarium* B-1. Genes annotated as hypothetical are coloured gray. Code and data are available at zenodo.org/records/15534318 ("notebook/figure2.ipynb" and "data/figure2/", respectively).

## Peptidoglycan hydrolases from Halogranum salarium B-1 kill a halophilic bacterium

Next, we sought to establish whether archaeal PGHs do indeed have antibacterial activity and, if so, which bacteria are targeted. To this end, we decided to focus on one of the chimeric zoocin A homologs we identified, from the halophilic archaeon *Halogranum salarium* B-1 (Uniprot ID: J2ZZK6, Fig 2C and 2D), which carries a signal peptide consistent with secretion by the Tat pathway (S2 Table).

As *H. salarium* B-1 was isolated from evaporitic salt crystals from the sea shore of Namhae, Korea, we name the J2ZZK6-encoding gene *woldo* ("moon blade"), in reference to a historical Korean polearm.

With an archaeal PGH homolog in hand, how can we identify the likely bacterial target(s)? We reasoned that, as the peptidoglycan-binding (PGB) domain of the PGH confers target specificity [38,39], we might home in on putative target bacteria by looking for homology between the PGB domain of a given archaeal PGH and PGB domains encoded in bacterial genomes, where they might be part of autolysins that need to act on the very same substrate as the archaeal PGH (Fig 3A). While perhaps not sufficient to confidently identify the bacterial target(s), this stratagem might nonetheless be helpful in whittling down a daunting multitude of potential targets to a priority list of candidate bacteria to test experimentally.

To explore whether this approach is likely to be informative, we first considered PGHs from phages, where they are known as endolysins. The advantage of phages is that we can predict their bacterial target with reasonable accuracy (or even certainty, in the case of temperate phage that have embedded themselves in the genome of their host, where they lie dormant until activated). We used the Virus–Host database [40] to map phages to their hosts, focusing on phages whose host is present in our genome database (S3 Table; see Methods). To avoid circularity, we removed all PGHs from the target database that are found in parts of bacterial genomes predicted to be prophages by PhiSpy [41]. We then asked how often, when carrying out homology searches with a given phage endolysin, we retrieve a PGH from the bacterial host as the top hit versus a PGH from another bacterium in the same database. Strikingly, the majority of host bacteria appear in the top 10 of the search results ($N = 256/479$, 53%) and the median rank of host bacteria is 1 (Fig 3B and S4 Table). Following the same approach, but for the catalytic domain of the phage endolysin, also regularly identifies the bacterial host in the top 10, but does so less consistently ($N = 170/479$, 35%; S2 Fig and S4 Table).

Encouraged by these results, we proceeded to predict bacterial targets of archaeal PGHs, using both protein sequence identity and structural similarity (based on RMSD) to assess similarity between bacterial and archaeal PGB domains (Fig 3A). Below, we focus on our focal archaeal protein (Woldo from *H. salarium* B-1), but predicted bacterial targets for the entire suite of archaeal PGHs, which are particularly enriched for Gram-positive bacteria from the phylum Bacillota (Fig 3E), can be found in S5 Table.

Considering the top bacterial hits for Woldo, we selected three halophilic bacteria for further investigation: *Halalkalibacterium halodurans* (UniProt ID of matched protein: Q9KDB8), *Virgibacillus salexigens* (UniProt: A0A024QGA1), and the moderately halophilic soil dweller *Phycicoccus endophyticus* (UniProt: A0A7G9R4I8, S6 Table and Fig 3D). We focus on these bacteria for three reasons: first, they are commercially available and can be cultured in isolation whereas many of the other top hits are against bacterial genomes assembled from metagenomic data (S6 Table); second, their basic

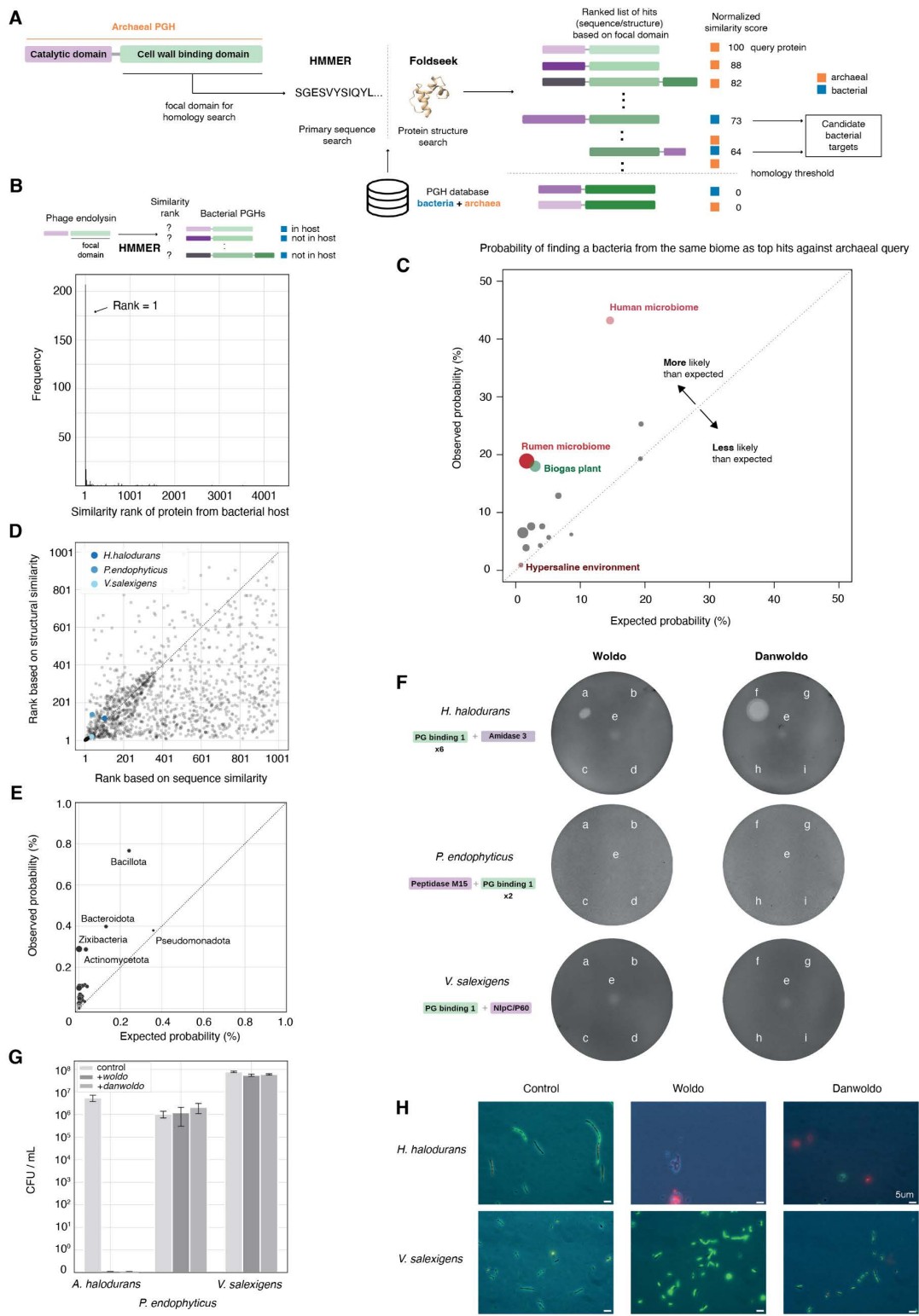

**Fig 3. Identification and validation of bacteria targeted by peptidoglycan hydrolases. A.** Schematic overview of the pipeline for identifying bacteria targeted by peptidoglycan hydrolases. The cell wall/peptidoglycan binding domain of a putative archaeal peptidoglycan hydrolase (PGH) is searched against a database of archaeal and bacterial PGH homologs, using either primary sequence (HMMer) or structural homology (Foldseek) searches.

Domain matches are ranked by similarity, normalized to the score obtained by matching against the query protein itself. **B.** The cell wall binding domains of phage endolysins are typically most similar to the cell wall binding domains of their (predicted) bacterial hosts compared to domains from other bacteria. **C.** Probability that, for a given archaeal query protein, bacteria from the same biome are identified amongst the top 10 hits. Biome labels were applied as described in Methods and results are aggregated at the biome level; the size of each circle indicates the likelihood ratio associated with a given biome. **D.** Ranking of homologous bacterial hits against the PG binding 1 domain of Woldo based on sequence vs. structural homology scores (Spearman's rho = 0.48). **E.** Enrichment of bacterial phyla amongst top 10 predicted targets across PGH homologs. See Methods for how enrichment was computed **F.** Results of spotting various supernatants onto lawns of *Halalkalibacterium halodurans, Phycicoccus endophyticus,* and *Virgibacillus salexigens.* a,b: top fraction (>3 kDa) of supernatant from *Haloferax volcanii* expressing an intact (a) or catalytic mutant (b) version of Woldo; c,d: bottom fraction (<3 kDa) of supernatant from *H. volcanii* expressing an intact (c) or catalytic mutant (d) version of Woldo; e: unfiltered supernatant from *Halogranum salarium* B-1; f,g: top fraction (>3 kDa) of supernatant from *H. volcanii* expressing (f) Danwoldo or (g) and empty plasmid control; h,i: bottom fraction (<3 kDa) of supernatant from *H. volcanii* expressing (h) Danwoldo or (i) and empty plasmid control. **G.** Colony forming assays for different bacteria following treatment with supernatant from *H. volcanii* expressing Woldo, Danwoldo, or the control (bottom fraction of Woldo-expressing *H. volcanii* H1424 supernatant). Error bars represent standard deviation from three biological replicates. **H.** Images of bacterial cells following exposure to *H. volcanii* supernatants expressing Woldo, Danwoldo or the control (as above). Bacteria are stained using LIVE/DEAD BacLight bacterial viability staining kit with ingress of red dye into bacterial cells indicative of cells with terminally compromised cell wall integrity. Code and data are available at zenodo.org/records/15534318 ("notebook/figure3.ipynb" and "data/figure3/", respectively).

growth requirements overlap with those of *H. salarium.* Even though the bacteria do not thrive at salinities that are optimal for *H. salarium*—2.6–3.4M NaCl [42]—the archaeon can survive and grow, albeit poorly, at the lower salinities (down to 1.4M NaCl) preferred by the bacteria. Consequently, these organisms might conceivably encounter each other in their natural niches; third, given that *H. salarium* is adapted to thrive under high salt conditions, it is reasonable to assume that its proteins, including Woldo, are adapted to function in a high-salt environment and might fail to function under low-salt conditions. Testing PGH functionality on bacteria growing in a halophilic medium reduces the risk that media composition impairs enzyme activity.

To test for PGH-mediated inhibition of putative target bacteria, we cloned *woldo* and expressed it in the model halophilic archaeon *Haloferax volcanii* (strain H1424, see Methods). Supernatant from *H. volcanii* expressing Woldo did not affect the growth of *P. endophyticus* or *V. salexigens* (Fig 3F). In contrast, we observed robust effects on the growth of *H. halodurans* (Figs 3F–3H and S3). Importantly, supernatant from *H. volcanii* without the expression plasmid or with a plasmid that carries a mutant version of Woldo with a compromised catalytic domain (see Methods) did not cause inhibition, suggesting that Woldo is the cause of, and catalytic activity critical for, inhibition.

*Streptococcus zoocin A* fatally compromises cell wall integrity and therefore has a bactericidal rather than bacteriostatic effect [43]. To establish whether this is also the case for Woldo, we carried out two complementary tests: colony forming assays and fluorescent live/dead staining. Both tests indicate bactericidal activity against *H. halodurans* (Figs 3G, 3H and S4).

### Bactericidal activity of Halogranum salarium supernatant

Is Woldo secreted by *H. salarium,* and does *H. salarium* supernatant inhibit bacterial growth? To address these questions, we spotted filtered *H. salarium* stationary phase (day 6) supernatant onto lawns of *H. halodurans*, *P. endophyticus*, and *V. salexigens*. *H. salarium* supernatant exhibited consistent bactericidal activity against *H. halodurans* (Fig 3F). However, bacteria often carry not one but several proteins that can be deployed against competitors. It would therefore be prudent to assume that Woldo is not the only protein in the *H. salarium* supernatant with bactericidal activity and that the native supernatant assay, therefore, does not directly implicate Woldo. Indeed, our PGH survey shows that *H. salarium* encodes a second PGH (UniProt ID: J2ZJU5, Fig 2B), composed of a glucosaminidase domain (PF01832) and two tandem PG binding 1 domains (PF01471) closely related to those in Woldo (Fig 2B). We name this protein Danwoldo, in reference to another Korean polearm, which is similar to Woldo but has a larger blade. Like Woldo, Danwoldo carries a Tat signal peptide consistent with active secretion.

To establish whether Woldo and/or Danwoldo are present in the bactericidal supernatant of *H. salarium*, we carried out quantitative label-free proteomics experiments on supernatant from stationary phase (day 6) *H. salarium* cultures. The 1,133 proteins we detect (S7 Table) include both Woldo and Danwoldo, with the latter more abundant than the former. To test whether Danwoldo too exhibits bactericidal activity, we carried out the same suite of experiments described above. We find that Danwoldo does indeed kill *H. halodurans* (but not the other tested bacteria) when secreted from *H. volcanii* H1424, and appears more potent than Woldo under the same conditions (Figs 3F–3H and S3). Danwoldo might therefore be partly (or even chiefly) responsible for the bactericidal activity of the native supernatant.

To confirm that the bactericidal activity of Danwoldo was not dependent on other factors in the *H. volcanii* supernatant, we fractionated the heterologous supernatant, identified killing fractions, and determined which proteins were present in those fractions using quantitative mass spectrometry (see Methods). Along with some proteins that are so abundant that some of their peptides are found across most fractions analyzed (pilA, petA, coxB1), Danwoldo is one the most abundant proteins in the killing fractions. More importantly, its abundance profile across neighboring fractions matches the relative bactericidal activity of those fractions (S3 Fig). This further strengthens our conclusion that Danwoldo kills bacteria without the need for other proteins.

Interestingly, *H. salarium* supernatant also mildly affected the growth of *V. salexigens*, despite the fact that neither Woldo nor Danwoldo, heterologously expressed from *H. volcanii*, exhibited bactericidal activity against this target (Fig 3F). This might suggest that successful targeting of this bacterium by Woldo/Danwoldo depends on additional factors not present in the supernatant of the heterologous expression system. Alternatively, *H. salarium* might secrete yet more factors, unrelated to the two PGHs, that compromise *V. salexigens* growth.

### Does the PGB homology approach identify ecologically relevant targets?

The observations above suggest that homology searches against the PGB domain(s) of archaeal PGHs might be useful to identify bacterial targets. Although further experimental work will be required to establish the specificity and sensitivity of this approach, we wondered whether the predicted targets broadly make ecological sense.

To address this question, we made use of the MGnify database, a large compilation of consistently labeled microbiome (meta)genomes [44]. We first assigned appropriate biome labels from the MGnify ontology to our original list of PGH-encoding archaea (see Methods). We then searched for bacterial homologs of archaeal PGHs in MGnify, asking whether, for a given archaeal PGH, the top-ranked bacterial hits share the same biome label as the producer more often than expected by chance (see Methods).

For 12 out of 14 biomes, we find evidence for non-random biome correspondence (likelihood ratio > 1, S8 Table and Fig 3C). Interestingly, bacteria from hypersaline environments are not significantly enriched amongst the top targets of archaeal hypersaline PGHs. On closer scrutiny, however, we find that top hits often come from biomes that could be considered frontier environments for halophiles, such as "fermented vegetables", which—as exemplified by kimchi—can also constitute salt-rich environments. *H. halodurans* itself is assigned the label "Environmental:Aquatic:Marine:Intertidal zone", the transitional zone where the ocean meets the land between high and low tides. As bacteria tend to increase in relative abundance compared to archaea as salinity drops [1], we propose that PGHs and similar antagonistic tools might be particularly useful in these environments.

### Discussion

We demonstrate above that a diverse assortment of archaea encode PGHs; enzymes that target a cellular structure, peptidoglycan, not found in archaea. At least three archaeal PGHs—Woldo and Danwoldo from *H. salarium* B-1, as shown here, and B5ID12 from *A. boonei* [30]—kill bacteria.

To identify putative bacterial targets of archaeal PGHs we used a (structural) homology approach, which—pending broader validation—might become a valuable tool for identifying producer-target relationships. Approaches of this

kind might be particularly useful for antagonistic interactions, where, for obvious reasons, producer and target are less likely to be found in the same sample, precluding some other common metrics to infer interactions (e.g., recurrent local co-occurrence, as one might expect under syntrophy).

Our work raises a number of questions. Arguably the most intriguing one is how archaeal PGHs (and potentially other archaeal proteins with antibacterial activity) are deployed in an ecological context. Woldo and Danwoldo are present in the supernatant of stationary phase *H. salarium* grown in monoculture, suggesting that they are expressed pre-emptively in nutrient-poor conditions, perhaps in anticipation of more competitive times ahead. Similar expression dynamics have previously been observed for halocins—small proteins that are secreted by halophilic archaea to kill other halophilic archaea [45].

But do archaea express PGHs to defend their niche and compete for resources? Or are bacteria themselves the resource, parcels of nutrition that are unlocked by PGHs? Is PGH activity principally directed at living bacteria (predation/ necrotrophy) or did it evolve for scavenging environmental peptidoglycan? Our results suggest that PGHs can kill *living* bacteria, but does not rule out scavenging as a major use of PGHs.

Establishing the adaptive value of different PGHs, and understanding what drove their acquisition and persistence, will require substantial future work, including in particular co-culture experiments, which can help establish if PGH-expressing archaea benefit nutritionally from killing co-habiting bacteria, if expression of PGHs is specifically induced by the presence of bacteria, and if specific bacteria are sensed specifically. It is interesting to note in this regard that genomes that encode a PGH are more likely than genomes without a PGH (23% versus 8%, Fisher test $P < 1.756^{-8}$) to encode a full set of homologs for the OppBCDF-MppA peptidoglycan uptake pathway, suggesting that, in at least some instances, peptidoglycan is taken up—and presumably utilized—by the archaeal cell.

Whatever the ecological drivers of PGH deployment, understanding how archaea interact with bacteria—through PGHs and other means—will be key to understanding how archaea persist in and shape polymicrobial ecosystems and to predicting short-and long-term change in the composition of these consortia. Because of their unique metabolic capacities, some archaea likely act as keystone species [4], with an outsize influence over community dynamics. Recognizing that archaea can interact in an antagonistic manner with bacteria in their community, actively shaping their niche, will change our appreciation of ecosystem function at equilibrium but also our ability to predict how interventions would play out.

## A roadmap for the discovery of additional archaeal-bacterial interactions

Our findings suggest, as much by implication as directly, that antagonistic interactions between archaea and bacteria might be pervasive. We think it unlikely that PGHs will turn out to be the sole mediators of archaeal-bacterial antagonism. Bacteria use a large variety of molecular systems to interfere with each other's growth and survival. The pan-bacterial arsenal includes large, complex molecular machines such as type VI secretion systems, contact dependent inhibition, bacteriocins that can penetrate and depolarize bacterial membranes, and an assortment of small molecules, that we have come to rely on as antibiotics [46]. Might archaea have access to an equivalent armory to kill bacteria? And how similar is this armory to that of bacteria?

We suggest four broad strategies to discover additional interactions in the future:

First, peptidoglycan is not the only biological structure that differentiates archaea from bacteria. Other acute points of difference exist, one being the lipids used in archaeal and bacterial membranes: archaeal lipids are made from isoprenoid chains that are linked by ether bonds to a glycerol-1-phosphate backbone. Bacteria, in contrast, predominantly use ester bonds to link fatty acids to a glycerol-3-phosphate backbone [47]. One might therefore pursue a similar strategy to the one we adopted above and search archaeal genomes for enzymes that specifically interact with bacterial phospholipids.

Second, whereas, at present, we do not know the potentially diverse methods by which archaea kill bacteria, we do know a great deal about how bacteria kill bacteria [46]. It would be interesting to establish systematically not only whether systems homologous to known bacterial weaponry are present in archaeal genomes [48] but also whether these are used

against bacteria. For example, it is known that many archaea encode contractile injection systems [48] and that these are used to kill other archaea [24], but whether the same systems target bacteria remains to be tested.

For a first glimpse into whether archaea might use the same toolkit as bacteria to kill bacteria we searched our archaeal genome collection for homologs of bacteriocins collated in the BAGEL4 database [49]. We find that some well-studied bacteriocins, like Lactoccocin 972, are never found in archaea (Fig 4). Others, like colicin M or sakacin A, are found in archaea in a patchy phylogenetic pattern reminiscent of their distribution in bacteria (Fig 4 and S9 Table) [28,50] and might therefore be good candidates for archaea-produced bacteriocins. However, candidate status here is less cogent than it is for PGHs and needs to be interpreted with care. Importantly, the link to bacteria is less specific: proteins as well as small metabolites could be active against bacteria but evolved to fight against other archaea, or for some other purpose altogether. To what extent these bacteriocin homologs are used by archaea to target bacteria will have to be established experimentally. In this context, it is worth pointing out a recent discovery of predicted pore-forming toxins (along with

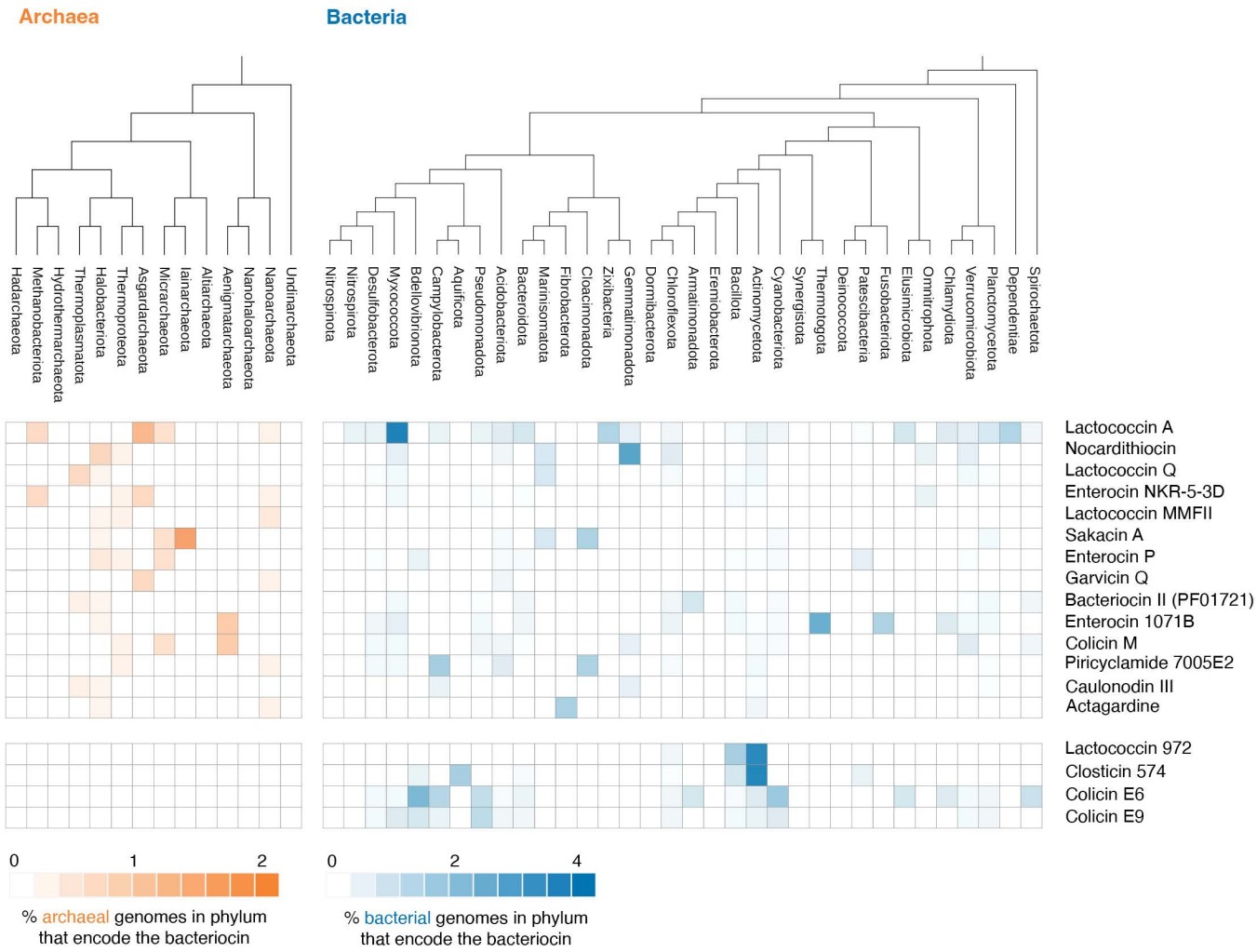

**Fig 4. Distribution of bacteriocin/archaeocin homologs across bacteria and archaea.** The relative abundance of different bacteriocin/archaeocin homologs is shown aggregated at the phylum level. Code and data are available at zenodo.org/records/15534318 ("notebook/figure4.ipynb" and "data/figure4/", respectively).

murein transglycosylases) in metagenome assembled genomes from the phylum Woesearchaeota, which also points to antagonistic interactions with bacteria [51].

Third, the two strategies above rely on homology to known bacterial systems as a starting point. These approaches, by themselves, will not reveal novel systems, especially systems unique to archaea. They may, however, provide critical seeds for discovery. A pertinent example here is the recent avalanche-like progress in identifying systems involved in conflicts between phage and bacteria: Genes involved in phage defense often cluster in bacterial genomes. Genes of unknown function that repeatedly co-occur with known defense genes were therefore predicted to be involved in anti-phage defense themselves—a guilt-by-association hypothesis that led to the discovery of multiple new phage defense systems [52]. As yet, we have too few landmarks (archaeal proteins with antibacterial function) to carry out similar analyses. As the number of landmarks grow, this might become a fruitful avenue for investigation. Provided, of course, that genes involved in antibacterial activity also have a tendency to cluster, which is by no means guaranteed.

Finally, bottom-up experimental approaches, including co-culture experiments, will remain invaluable to enable discovery of novel, uniquely archaeal proteins and small molecules with antibacterial activity, to provide the foundations for guilt-by-association approaches, to understand how and to what effect these tools are deployed against bacterial targets, and to assess whether novel archaeal antibacterials might be promising leads for future clinical applications.

## Methods

### Prokaryotic genome database

We assembled a database of 54,346 prokaryotic genomes (50,640 bacteria and 3,706 archaea), constructed as a phylogenetically balanced subset of release 214 of GTDB, the Genome Taxonomy Database [53], selecting a single representative strain per species (as per the representative strain flag available in GTDB) and a maximum of 10 species per genus, prioritizing those species with most strains. A full list of genomes and corresponding assemblies is provided in S1 Table. Note that, when choosing representative strains, GTDB prioritizes genomes from type strains and of high assembly quality, so that our database is weighted towards higher-quality genomes [53,54].

### Identifying peptidoglycan hydrolases

We define PGHs as proteins made of at least one Pfam catalytic domain known to cleave peptidoglycan, and at least one Pfam cell wall-binding domain, cross-referencing our manually curated list with literature on bacterial autolysins [55]. Manual curation was based on a domain's Pfam description: all domains mentioning peptidoglycan hydrolase activity or peptidoglycan binding activity have been included. A full list of qualifying Pfam domains is available as S10 Table. Proteins matching this definition were found by systematically searching our database of prokaryotic genomes using Pfam's Hidden Markov Models (HMMs) with the HMMER software suite (hmmer.org). Signal peptides were predicted with SignalP 6 [56]. The full list of proteins found is available as S2 Table.

PGHs were assigned to main chromosome, plasmid, contig, etc. as follows. For complete genomes, existing annotations were used to assign the PGH to a chromosomal or plasmid location. For incomplete genomes, the size of the contig was compared to the median contig size of the assembly. Contigs below the median size were considered "small contigs". Scaffold level genomes were excluded from this analysis.

### Phylogenetic trees

Cladograms (Figs 1C and 4) are derived from the bacterial and archaeal trees from GTDB: For visualization purposes, the cladograms depicted in Figs 1C and 4 are limited to named phyla, i.e., excluding automatically generated phylum names composed of numbers and letters. Phyla with the same name but differentiated by a letter (such as Pseudomonata, Pseudomonata_A, Pseudomonata_B, etc.) were grouped into a single phylum (e.g., Pseudomonata). Hits against *all* phyla (whether named or not) are found in S2 Table.

Protein domain trees (Fig 2) were constructed by selecting the top 1,000 hits to the peptidase M23 and PG binding 1 domains of protein J2ZZK6 (Woldo), respectively. Sequences were aligned with MAFFT using the L-INS-I strategy [57]. Alignment positions with less than 35% aligned residues [58] were trimmed with trimAl [59]. Maximum likelihood trees were built with IQ-TREE 2 [60] with automatic model finder ("-m MFP" option) and UFBoot2 approximate bootstrap values ("-B 1000" option). Only the bacterial clades branching with archaea plus one outgroup were kept for clarity. Alignments and tree files are available at zenodo.org/records/15534318 (folder "data/figure2"). The tree in S1 Fig is a subset of the GTDB archaeal tree containing only Halobacteriota species present in our database. All trees were rendered with iTOL [61].

### Phage–Host database

Bacteriophages and their host were retrieved from the Virus–Host database [40]. Among these bacteriophages, only the ones whose host belongs to our genome database were kept, using the bacterium's NCBI taxonomy ID to perform the matching (S3 Table). Whenever a given NCBI taxonomic ID points to >1 GTDB genome in our database, we pick the GTDB representative genome ("gtdb_type_species_of_genus" in GTDB) if available or else the genome with the highest completeness ("checkm_completeness" in GTDB). We then considered phage genomes that contained at least one endolysin.

### Predicting bacterial targets of archaeal PGHs and phage endolysins

To identify the likely bacterial targets of phage or archaeal PGHs, we used HMMER (phmmer) to carry out homology searches against our prokaryotic database, using only the cell wall binding domain as a query and keeping proteins with an $E$-value ≤ 1e−6. Following prediction of tertiary structures of the 1,372 full-length homologous target proteins using ColabFold (v1.5.2) [62], target proteins were ranked either by protein sequence identity or structural similarity (measured as RMSD) to the archaeal query domain using the HMMER and Foldseek output [63], respectively (Fig 3D).

### Biome predictions

To evaluate how often our target search approach identifies bacteria that inhabit the same or a similar environment to the archaeal PGH producer, suggestive of ecologically relevant cross-domain interactions, we require a set of consistent biome labels across bacterial and archaeal genomes. Controlled labeling of this type has been done for genomes in the MGnify database [44]. We matched proteins from this dataset to the PGH proteins in our database using phmmer with an $E$-value threshold of 1e−6, in effect importing labels from MGnify into our database using homology as a guide. Many of the MGnify biome labels are broad (e.g., Environmental:Aquatic:Marine). Therefore, we picked the three top biome labels per PGH protein. For each archaeal PGH protein, we extracted the biome labels of the top 10 predicted bacterial targets and asked whether the probability of the archaeal producer label to appear in the top 10 bacterial targets is statistically different from randomly sampling 10 labels from the entire bacterial PGH dataset.

### Phylum enrichment

To establish whether particular phyla were enriched amongst predicted bacterial PGH targets, we considered the entire set of bacterial PGHs in our database (S2 Table) and computed the probability of retrieving a hit from a particular phylum by chance and multiplied this number by 10 to generate an expected probability of a protein to appear amongst the top 10 hits. We then considered the bacterial PGH hits for each archaeal PGH (S5 Table) and computed the observed probability of a phylum to appear in the top 10. Details can be found at zenodo.org/records/15534318 (file "notebook/figureS2.ipynb").

### Antimicrobial proteins shared between archaea and bacteria

To identify putative antibacterial proteins in archaea we searched in our database for homologs to antimicrobial proteins present in the BAGEL4 database [49]. Proteins in BAGEL4 were clustered using CD-HIT [64] to group together proteins

identical or similar in sequence (90% similarity; option -c 0.9) and this consolidated set of proteins were used as queries to search against our database using HMMER, with an *E*-value threshold of 1e−6 (S9 Table). The heatmap in Fig 4 only includes proteins present in both archaea and bacteria in at least two genomes and two phyla in each domain, plus four selected bacteriocins not present in archaea. The set shared by archaea and bacteria was further reduced to exclude the following proteins labeled as bacteriocins in BAGEL4: PGHs; Linocin M18, which has known non-bactericidal functions in *Pyrococcus furiosus* [65]; FlvA2f and FlvA2h, which have been shown to have no bactericidal activity [66]; and comX homologs, pheromones involved in quorum sensing [67], since no bactericidal activity has been described.

### Strains and culture conditions

*Halogranum salarium* B-1 (DSM-23171), *Halalkalibacterium halodurans* (DSM-18197), *Phycicoccus endophyticus* (DSM-100020), and *Virgibacillus salexigens* (DSM-11483) were obtained from the Deutsche Sammlung von Mikroorganismen und Zellkulturen (DSMZ). *Haloferax volcanii* H1424 was a gift from Thorsten Allers (University of Nottingham). *H. salarium* B-1 was grown in DSMZ media recipe 1,377 (13% w/v NaCl). *H. volcanii* was grown in medium Hv-YPC 18% w/v NaCl as previously described [68]. The three bacteria *H. halodurans*, *P. endophyticus,* and *V. salexigens* were grown in Hv-YPC 8% w/v NaCl. Note here that *Halogranum salarium* B-1 can also grow, albeit not optimally, at 8% w/v NaCl. Pre-cultures from −80 °C glycerol stocks were grown in 3 mL volume in 15 mL Falcon tubes. Cultures were grown in 20 mL volume in 100 mL Erlenmeyer flasks. *H. salarium B-1*, *H. halodurans*, *P. endophyticus*, and *V. salexigens* were grown at 37 °C, while *H. volcanii* was grown at 45 °C, all in a shaking incubator at 180 rpm.

### Transformation of Haloferax volcanii

DNA sequences corresponding to J2ZZK6 (*woldo*) and J2ZJU5 (*danwoldo*) were amplified from *H. salarium* B-1 genomic DNA by PCR (see S11 Table for primers) and cloned into *H. volcanii* plasmid pTA1392 (a gift from Thorsten Allers) as follows: the plasmid was digested with NdeI and BamHI and subsequently dephosphorylated using the shrimp alkaline phosphatase (New England Biolabs, NEB). The digested PCR product (*woldo* or *danwoldo*) and the dephosphorylated vector were ligated using T4 DNA ligase (NEB). The ligation mixture was electroporated into *Escherichia coli DH5a*, and transformed clones were selected on LB agar + 100 µg/mL carbenicillin. The resulting plasmids were extracted using the Monarch Plasmid miniprep kit (NEB) and transformed into *H. volcanii* H1424 by following an established transformation protocol [68]. To serve as a control, the original pTA1392 without a gene insert was similarly transformed into *H. volcanii* H1424 using the same protocol. Finally, we sought to construct a catalytic mutant of Woldo. Introducing a point mutation via PCR proved difficult because the high GC content of *woldo* prevented design of workable primers. Instead, we synthetized a suitable DNA fragment containing the necessary mutation and attempted to introduce the fragment into pTA1392 via Gibson assembly. None of the mutants we screened had a successful integration event. However, we fortuitously isolated a mutant plasmid with deletion that covered the catalytic center of J2ZZK6 (Δ91−149). All plasmids were sequenced (via Plasmidsaurus) prior to deployment to confirm expected integration events.

### Inhibition assays

*H. salarium* cultures were inoculated at 0.01 $OD_{600}$ from overnight cultures, into 20 mL 13% w/v NaCl media in 100 mL Erlenmeyer flasks and grown for 6 days. *H. volcanii* cultures were inoculated at 0.01 $OD_{600}$ from overnight cultures, into 20 mL Hv-YPC 13% w/v NaCl + 3 mM Tryptophan to induce expression, in 100 mL Erlenmeyer flasks overnight.

Supernatant was extracted by centrifugation of liquid cultures of *H. salarium* or *H. volcanii* at 4,000*g* for 10 min. Supernatant was subsequently passed through 0.2 µm Minisart PES sterile filters to remove cells. Supernatant was then concentrated using 3,000 MWCO PES membranes (Vivaspin 20) by centrifugation for 45 min at 4,000*g*. Volume in the top compartment was reduced from 20 to 1 mL (20×) and is enriched in molecules with a molecular weight >3 kDa. The bottom compartment holds the flow through. Spotting assays on target lawns were carried as previously [69]. Briefly, target

bacterial cells were incorporated into a soft agar top layer, which was poured hot into a plate containing hard agar of the bacterial medium. Supernatant was spotted on top of the plate straight after the top layer had solidified. Plates were left for colonies to grow at 37 °C and imaged after 24 h and inspected for clearings.

### Colony-forming units

Bacterial cultures of *H. halodurans*, *P. endophyticus*, and *V. salexigens* were grown as described above to an $OD_{600}$ of 0.5. Concentrated supernatant containing Woldo and Danwoldo, respectively, were collected as described above. The flow through from supernatant concentration of J2ZZK6 was used as control. A volume of 1 µL of bacterial culture was added to 9 µL of supernatant in Starlab PCR tubes, for a total of 18 tubes (3 bacteria, 3 conditions, and 2 replicates). The mixtures were incubated for 1 hour at 37 °C in a shaking incubator at 180 rpm. Two 1:100 serial dilutions were performed in bacterial growth media bringing the final dilution factor to 10,000. A volume of 100 µL was plated on pre-warmed solidified bacterial media. Plates were incubated for 36 h at 37 °C and imaged and colonies counted using ImageJ (protocol: 10.17504/protocols.io.f2mbqc6).

### Imaging and live/dead staining

Bacterial cultures of *H. halodurans* and *V. salexigens* were grown as described above to an $OD_{600}$ of 0.5. 1 mL of culture was centrifuged for 10 min at 4,000$g$. The pellet was resuspended in 500 µL of culture media. Concentrated supernatant from cultures expressing Woldo and Danwoldo, respectively, were collected as described above. The flow through from supernatant concentration of J2ZZK6 was used as control. A volume of 1 µL of bacterial culture was added to 9 µL of supernatant in Starlab PCR tubes, for a total of six tubes (two bacteria, three conditions). The mixtures were incubated for 1 h at 37 °C in a shaking incubator at 180 rpm. Dyes and protocols from LIVE/DEAD BacLight bacterial viability kits were used. Briefly, equal volumes of SYTO 9 dye (3.34 mM) and Propidium iodide (20 mM) were mixed. The mixture was diluted 1:10 in deionized water. A volume of 0.3 µL was added to each of the six tubes and incubated in the dark at room temperature for 15 min. A volume of 3 µL of the stained bacterial suspension was trapped between a microscopy slide and a coverslip. Observations were carried out using a Leica DMRB upright microscope with a Leica 100×/1.30 Oil PL Fluotar objective. We used two Semrock Penta florescence filters (FITC 485/20, TRITC 560/25). Acquisition was performed with the Hamamatsu Orca camera using software MicroManager [70] with 10 ms exposure for the fluorescence channels and 25 ms for the brightfield channel. Composite images were assembled with ImageJ.

### Liquid chromatography–tandem mass spectrometry (LC–MS) sample preparation—Native supernatant

Supernatant was extracted from 20 mL cultures of *H. salarium* grown in 100 mL Erlenmeyer flasks for 6 days, in biological triplicates. Secreted proteins were purified following chloroform-methanol extraction in the presence of protease inhibitor (cOmplete protease inhibitor cocktail from Sigma). For ease of manipulation, several 200 µL aliquots were processed in parallel in 2 mL Eppendorf tubes. Each aliquot was mixed with 800 µL methanol, 200 µL chloroform, 600 µL of purified water, and vortexed until solution turns white. Precipitated proteins were centrifugated at 17,000$g$ for 10 min. Next, aqueous supernatant was removed while being careful not to disturb the white protein pellet. Finally, 860 µL of methanol were added and the solution mixed gently. Following 5 min of centrifugation at 17,000$g$, the chloroform methanol mix was discarded and the pellet allowed to dry for up to 5 min. 100 µg of the proteins obtained were processed into purified peptides using the PreOmics iST kit, following manufacturer's instruction, including a 1.5 h digestion step. Purified peptides were stored in LOAD buffer at −80 °C until injection.

### Liquid chromatography–tandem mass spectrometry (LC–MS) sample preparation—Heterologous supernatant

Approximately 100 mL of media were concentrated to 1 mL using a 3 kDA MWCO filter. This was then injected on SEC column and fractionated into 1 ml fractions. Fourteen fractions were analyzed: five fractions before the fraction

with the most antimicrobial activity and eight fractions after. 200 µl of every fraction was processed. pH was adjusted to 8.5 with 50 mM EPPS, the samples were then reduced and alkylated with 10 mM TCEP and 20 mM cholracetamide for 30 min, and then digested with 10 ng/µl trypsin (Pierce P/N 90059) and 50 ng/µL LysC (WAKO) shaking at 300 rpm in a 500 µL low-binding Eppendorf plate. The digest was desalted using 3 mg oasis HLB 30 µm resin packed in Orochem OF1100 filter plate. Samples were resuspended in 30 µl of 0.1% TFA and 5 µl of sample was analyzed by LC–MS

## LC–MS acquisition—Native supernatant

Samples were injected and data acquired in single replicate injections as follows:

Chromatographic separation was performed using an Ultimate 3000 RSLC nano liquid chromatography system (Thermo Scientific) coupled to an Orbitrap Exploris 240 mass spectrometer (Thermo Scientific) via an EASY-Spray source. Peptide solutions were injected directly onto the analytical column (Self-packed column, CSH C18 1.7 µm beads, 150 µm × 35 cm) at working flow rate of 1.3 µL/min for 8 min. Peptides were then separated using a 69-min stepped gradient: 0%–25% of buffer B for 49 min, 25%–42% of buffer B for 20 min (composition of buffer A—95/5%: $H_2O$/DMSO + 0.1% FA, buffer B—75/20/5% MeCN/$H_2O$/DMSO + 0.1% FA), followed by column conditioning and equilibration. Eluted peptides were analyzed by the mass spectrometer in positive polarity using a data-independent acquisition mode as follows: an initial MS1 scan was carried out at 120,000 resolution for 25 ms in profile mode, $m/z$ range: 409.5–1,650 and normalized AGC target: 300%. This was followed by sequential MS2 acquisition and fragmentation of ions at 30,000 resolutions in centroid mode over 30 variable windows, $m/z$ range: 145–1,450, normalized AGC target: 2,000% and normalized collision energy: 27%. Total run acquisition time was 84 min.

## LC–MS acquisition—Heterologous supernatant

Chromatographic separation was performed using an Ultimate 3000 RSLC nano liquid chromatography system (Thermo Scientific) coupled to an Orbitrap exploris 240 mass spectrometer (Thermo Scientific) via an EASY-Spray source. Electrospray nebulization achieved by interfacing to Bruker PepSep emitters (PN: PSFSELJ20, 20µm). Peptide solutions were injected directly onto the analytical column (self-packed column, CSH C18 1.7µm beads, 300µm × 30 cm) at working flow rate of 5 µL/min for 4 min. Peptides were then separated using a 60 min segmented gradient: 0%–42% of buffer B for 60 min (composition of buffer A—95/5%: $H_2O$/DMSO + 0.1% FA, buffer B—75/20/5% MeCN/$H_2O$/DMSO + 0.1% FA), followed by column conditioning and equilibration. Eluted peptides were analyzed by the mass spectrometer in positive polarity using a data-independent acquisition mode as follows: an initial MS1 scan was carried out at 120,000 resolution with a normalized AGC target of 300% for a maximum IT of 60 ms, m/z range: 409.5–1,600. This was followed by a 120,000 boxcar scan of 10 segments each injected for a maximum of 25 msec and normalized AGC of 100%. This was followed with a sequential MS2 acquisition and fragmentation of ions at 30,000 resolution over 30 variable windows. Normalized collision energy was set to 27%. Total run acquisition time was 70 min.

## LC–MS data processing—Native supernatant

Data were processed using the Spectronaut software platform (Biognosys, v17.7.230531) [71]. Analysis was carried out in direct DIA mode as follows:

1. Pulsar Search: library generation and database searching were carried out using default settings for a tryspin/p--specific digest as follows and with the following adjustments–missed cleavage rate set to 3 and variable modifications allowed for methionine oxidation, protein N-terminal acetylation, asparagine deamidation and cyclization of glutamine to pyro-glutamate. PSM, Peptide and Protein group FDR = 0.01. Searches were run against a user-generated protein sequence database for *H. salarium* B-1 (RefSeq: GCF_000283335.1).

2. *DIA analysis*: a mutated decoy database approach was employed with protein *q*-value cut-off for the experiment set to 0.05 at the identification level. Quantification set to MS2 with proteotypicity filter set to only proteotypic peptides with no value imputation strategy employed. Protein quantification method set to MaxLFQ [72].

### LC–MS data processing—Heterologous supernatant

Data were processed using the Spectronaut software platform (Biognosys, v 19.9.250512.62635). Pulsar search was performed with default settings for a tryspin/p-specific digest with missed cleavage rate set to 3 and a fixed modification of cysteine carbamidomethylation. Variable modifications allowed for methionine oxidation, protein N-terminal acetylation, and cyclization of glutamine to pyro-glutamate. Searches were carried out against a reference *Haloferax volcanii* D2 protein sequence database, supplemented with the Danwoldo protein sequence (UniProt: J2ZJU5) and a universal protein contaminants database (downloaded 22/01/2024, 381 entries) [73]. A mutated decoy database approach was employed with protein PEP cut-off for the experiment set to 0.01 at the identification level. Quantification was performed at MS2 level with no value imputation strategy employed. Protein and peptide level cross-normalization was disabled, since we did not expect equal total signal in every SEC fraction.

### Protein purification using gel filtration

Concentrated supernatant (25×) from *H. volcanii* strains expressing Danwoldo were prepared as described for the inhibition assay. Using an AKTA prime plus robot (GE Helthcare), a 1 mL sample was injected onto a HiLoad 16/600 Superdex 200 prep grad column (GE Healthcare) pre-equilibrated in saltwater buffer [18% w/v NaCl; as per [68] chapter 3, p 38] at room temperature. Elution was performed with the same buffer at a flow rate of 1 mL/min. Fractions of 1 mL were collected throughout the elution, and their bactericidal activity was assessed against *H. halodurans* following the inhibition assay protocol described above. Fractions exhibiting inhibition of *H. halodurans* growth were selected for SDS-PAGE analysis. Proteins were extracted by methanol-chloroform precipitation (as described in section *LC–MS sample preparation*) and resuspended in 30 μL of deionized water plus 10 μL NuPAGE LDS Sample Buffer (4×). Samples were heated for 10 min at 70 °C. Samples were then loaded on a Bolt Bis-Tris gel alongside 7 μL of molecular weight marker (Thermo Scientific 26619). Electrophoresis was carried out at 165V for 45 min. The gel was then stained using InstantBlue Coomassie Protein Stain (Abcam ab119211).

### Supporting information

**S1 Fig. Presence/absence of peptidoglycan hydrolase homologs across the phylum Halobacteriota.** The tree shown is based on the GTDB archaeal tree but pruned to only contain Halobacteriota species present in our database. Tree scale: substitutions per site. Code and data are available at zenodo.org/records/15534318 ("notebook/figureS1.ipynb" and "data/figureS1/", respectively).
(PDF)

**S2 Fig. Rank of the bacterial host peptidoglycan hydrolase catalytic domain when searching our bacterial database with the catalytic domain of a given phage endolysin.** This figure corresponds to Fig 3B but for the catalytic (CAT) instead of the cell wall binding domain. Code and data are available at zenodo.org/records/15534318 ("notebook/figure3.ipynb" and "data/figure3/", respectively).
(PDF)

**S3 Fig.** A. Comparing the inhibitory effects of Woldo and Danwoldo on the same plate. The following supernatants were spotted onto a lawn of *H. halodurans*, a,c,e: top fraction (>3 kDa) of supernatant from *H. volcanii* (a) expressing Woldo, (c) expressing Danwoldo, (e) with an empty plasmid; b,d, f: bottom fraction (<3 kDa) of supernatant from *H.*

*volcanii* (b) expressing Woldo, (d) expressing Danwoldo, (f) with an empty plasmid. B. Lawn of *H. halodurans* exposed to the cell-free supernatant (CFS) of danwoldo-expressing *H. volcanii*, and defined fractions of that supernatant (see [Methods]). C. Coomassie-stained SDS-PAGE gel highlighting the protein contents of peak killing fractions (56/57) and neighboring fractions. The expected size of Danwoldo is indicated by arrows. D. Protein abundance profile (based on IBAQ measurements from quantitative proteomics, see [Methods]) of Danwoldo across killing and neighboring fractions. Code and data are available at zenodo.org/records/15534318 ("notebook/figureS3.R" and "data/figureS3"), respectively.
(PDF)

**S4 Fig. Plates used to determine colony forming units, following exposure to supernatant containing Woldo or Danwoldo.**
(PDF)

**S1 Table. Genomes in the phylogenetically balanced database, including 3,706 archaeal and 50,640 bacterial genomes.**
(XLSX)

**S2 Table. Peptidoglycan hydrolases detected in bacteria and archaea.**
(XLSX)

**S3 Table. Phage-bacterial host assignments from on Virus–Host database for the bacterial genomes in our database.**
(XLSX)

**S4 Table. Correspondence between phage endolysin domains and homologous domains in the bacterial host.**
(XLSX)

**S5 Table. Top 10 predicted bacterial targets based on sequence homology for all archaeal peptidoglycan hydrolases.** This table is available at zenodo.org/records/15530213.
(ZIP)

**S6 Table. Domains homologous to Woldo and Danwoldo amongst archaea/bacteria in the database.**
(XLSX)

**S7 Table. Proteins identified in stationary phase (day 6) supernatant of *Halogranum salarium* B-1 cultures.**
(XLSX)

**S8 Table. Enrichment of MGnify biome labels between archaeal queries and their putative bacterial targets.**
(XLSX)

**S9 Table. Archaeal homologs of bacteriocins.**
(XLSX)

**S10 Table. List of domains belonging to putative peptidoglycan hydrolases.** Proteins containing at least one catalytic domain and at least one cell wall binding domain from this list were used for homology searches in archaeal genomes.
(XLSX)

**S11 Table. Primers used for the amplification of *woldo* and *danwoldo*.**
(XLSX)

## Acknowledgments

We thank Thorsten Allers for *H. volcanii* strains and plasmids, and Chahrazad Taissir and Nika Pende for help and advice.

## Author contributions

**Conceptualization:** Antoine Hocher, Tobias Warnecke.

**Data curation:** Romain Strock.

**Formal analysis:** Romain Strock, Pavel V Shliaha, Tobias Warnecke.

**Funding acquisition:** Tobias Warnecke.

**Investigation:** Romain Strock, Valerie WC Soo, Pauline Misson, Georgia Roumelioti, Pavel V Shliaha, Tobias Warnecke.

**Methodology:** Romain Strock, Valerie WC Soo, Pauline Misson, Pavel V Shliaha, Antoine Hocher, Tobias Warnecke.

**Project administration:** Romain Strock, Tobias Warnecke.

**Resources:** Valerie WC Soo.

**Supervision:** Antoine Hocher, Tobias Warnecke.

**Visualization:** Romain Strock, Tobias Warnecke.

**Writing – original draft:** Romain Strock, Valerie WC Soo, Pauline Misson, Pavel V Shliaha, Antoine Hocher, Tobias Warnecke.

**Writing – review & editing:** Romain Strock, Valerie WC Soo, Antoine Hocher, Tobias Warnecke.

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
