## [Editor Report · Decision Letter 0]

Dear Dr Warnecke, 

Thank you for submitting your manuscript entitled "Systematic genome-guided discovery of antagonistic interactions between archaea and bacteria" for consideration as a Research Article by PLOS Biology.

Your manuscript has now been evaluated by the PLOS Biology editorial staff, and I'm writing to let you know that we would like to send your submission out for external peer review. I should warn you that we were not able to secure advice from an Academic Editor in a timely fashion, so we'll be looking for some enthusiasm from the reviewers.

Once your full submission is complete, your paper will undergo a series of checks in preparation for peer review. After your manuscript has passed the checks it will be sent out for review. To provide the metadata for your submission, please Login to Editorial Manager (https://www.editorialmanager.com/pbiology) within two working days, i.e. by Dec 09 2024 11:59PM.

Kind regards,

Roli Roberts

Roland Roberts, PhD

Senior Editor

PLOS Biology

rroberts@plos.org

---

## [Decision Letter · Decision Letter 1]

Dear Toby,

Thank you for your patience while your manuscript "Systematic genome-guided discovery of antagonistic interactions between archaea and bacteria" went through peer-review at PLOS Biology. Your manuscript has now been evaluated by the PLOS Biology editors, an Academic Editor with relevant expertise, and by four independent reviewers. Please accept my further apologies for the time that this process has taken.

You'll see that reviewer #1 is very positive; their requests are mostly presentational, but do include one “simple BLASTP search.” Reviewer #2 is also broadly positive, but wants improvements to the logical flow, and sounds sceptical of the peptidase M23 tree. There are several potential analyses, and one wet-lab experiment (to use purified proteins for the hydrolysis validation, rather than crude lysates). Reviewer #3 likes it, but has a number of textual suggestions (including a change to the Title, as did rev #2); s/he also wants to know if you can show that the archaeal plasmids are actually in archaeal hosts (rather than bacterial ones), asks about presence of genes encoding agents for peptidoglycan uptake and processing, and wonders if the genes encoding peptidoglycan hydrolases lie with operons – some of these may require new analyses. Reviewer #4 has a number of textual requests, but there are a few simple analyses buried in the middle.

In light of the reviews, which you will find at the end of this email, we are pleased to offer you the opportunity to address the comments from the reviewers in a revision that we anticipate should not take you very long. We will then assess your revised manuscript and your response to the reviewers' comments with our Academic Editor aiming to avoid further rounds of peer-review, although we might need to consult with the reviewers, depending on the nature of the revisions.

**IMPORTANT - SUBMITTING YOUR REVISION**

*Resubmission Checklist*

*Published Peer Review*

*PLOS Data Policy*

*Blot and Gel Data Policy*

Sincerely,

Roli

Roland Roberts, PhD

Senior Editor

PLOS Biology

rroberts@plos.org

REVIEWERS' COMMENTS:

Reviewer #1:

General comments:

This is an exciting and important paper looking a much neglected field - natural antagonistic interactions between archaea and bacteria. The genomics-based observations are backed up by some proof-of-concept experiments in the lab, which are convincing. Nonetheless, I have some comments that should be considered.

Major comments:

Phylogeny-based conclusions rely on decent MSAs so that the trees really represent protein evolution. MSAs should therefore be provided in supplementary materials. Additionally the scale of the trees is not clear: generally a substitution per site scale is provided, but what is the scale here?

2/3 of archaeal PGH homologs do not encode a known signal peptide. What do the authors think about potential function of these proteins? The authors correctly point out that "pseudo-peptidoglycan (pseudomurein) exhibits different composition and linkage patterns, making it chemically distinct from bacterial peptidoglycan". However there are also many similarities, and some conclude that they have a shared ancestry (see doi: 10.1093/femsmc/xtab012). A simple BLASTP search would have established whether some of the species that have PGH homologs have a partial or complete pseudomurein biosynthesis cluster.

Minor comments:

The authors jump from phylum to order level in a way that is sometimes confusing, especially since some members of Halobacteriota are not halophiles.

The choice of J2ZZK6 was probably also influenced by the fact that it has a secretion signal (TAT), this should be mentioned.

the archaeon can survive and grow at the lower salinities (up to 1.4M NaCl) - should be "down to" or "as low as"

the risk that media composition scuppers enzyme activity - to the best of my knowledge to scupper means to sink your own ship on purpose (metaphorically), I think "impairs" will be more accurate and much clearer to a non UK readership.

at fault for the bactericidal activity of the native supernatant. - should be "responsible for"

"We speculate that, for these species, PGH might nonetheless be useful when they encounter what, for them, are frontier environments: niches at the edge of their range, where lower salinity allows bacteria to be competitive and archaea survive rather than thrive. In the future, it will be interesting to define the (stress) conditions that trigger PGH expression and how they relate to the ecophysiology of the particular archaeon under investigation."

The authors seem to ignore the possibility of necrotrophy - because bacteria that are not high salt adapted may end up in evaporation pools, for example, where they will dehydrate and die, this could provide a source of nutrition for haloarchaea able to secrete PGHs.

Type VI secretion-based antagonism - https://www.science.org/doi/10.1126/sciadv.adp7088 should be cited and referred to. 

Reviewer #2:

The manuscript, titled "Systematic genome-guided discovery of antagonistic interactions between archaea and bacteria", systematically presents the distribution of peptidoglycan hydrolases in archaea. These hydrolases target the peptidoglycan in the bacterial cell wall and exhibit bactericidal activity. This research offers valuable perspectives on the potential of archaea-derived antibacterials and the relatively under-explored antagonistic interactions between archaea and bacteria within mixed microbial communities. I recommend the publication of this paper, following necessary revisions.

Main Comments:

1. The study consistently focuses on peptidoglycan hydrolases through high-throughput genomic analysis and experimental validation. To more accurately encapsulate the essence of the research and distinguish it from other studies on antagonistic interactions, it would be beneficial to include "Peptidoglycan Hydrolases" in the title.

2. It is strongly recommended to clearly state the number of zoocin A homologs identified in archaeal genomes, as well as the distribution patterns of these analogs. Additionally, an explanation is needed regarding the rationale behind selecting only 10 of these homologs for the construction of the genetic tree analysis.

3. In the manuscript, the names of the proteins should be presented in the results section rather than in the methods section. This would improve the logical flow and make it easier for readers to follow the key findings related to the proteins.

4. Is it accurate that the RMSD value of the full-protein match is larger than that of the partial-protein match?

5. The protein-domain-level genetic tree of peptidase M23 reveals an unusual phenomenon: proteins from haloarchaea with similar domain structures are not clustered together. This unexpected result requires in-depth analysis and a more detailed explanation.

6. While the in-vitro hydrolysis validation experiment was performed using the crude extract of the heterologous expression and the supernatant of the wild-type strains, a more definitive conclusion could be drawn by repeating the experiment using pure protein and substrates, which is feasible.

7. Considering that the subsequent protease J2ZJU5 can be predicted in silico and is forecasted to possess a signal peptide, the author should provide a rationale for the continued use of proteomics analysis to guide the discovery process.

8. Figure 4, as it stands, appears to be disconnected from the overall logical structure of the manuscript. Its relevance should be reassessed, or alternatively, a clear link to the central argument should be established within the discussion section.

Reviewer #3:

This interesting manuscript reports the identification of peptidoglycan hydrolase genes in archaeal genomes and that the corresponding enzymes are presumably used to target bacteria in interspecies competition. The supplemental files with the genes/proteins identified will be a valuable source for the field. Unfortunately, the manuscript has not page/line numbers.

Main points:

1. I suggest changing the title, which claims that they identified antagonistic interactions between archaea and bacteria. However, this has not been tested. To test for antagonistic interactions, they would have to co-culture a target bacterium with a hydrolase-producing archaea in a competition assay that quantifies survival of both. At present, they can only conclude that some archaea secrete peptidoglycan hydrolases and two of these can kill a bacterium that might occupy the same environmental niche. In my view this is already an important result.

2. They searched for archaeal genes that encode for a peptidoglycan hydrolase domain and a peptidoglycan-binding domain. This combination of an enzymatic and a binding domain is mainly found in bacterial enzymes that target Gram-positive bacteria. There are many peptidoglycan hydrolases that lack a peptidoglycan binding-domain and target Gram-negative bacteria or even Gram-positives. A well-known example is lysozyme. Hence, they should clearly state that their strategy does not identify single-(enzymatic) domain hydrolases and could be biased towards hydrolases that target Gram-positive bacteria.

3. The introduction lacks the mentioning of peptidoglycan hydrolases secreted by type VI systems and targeting Gram-negatives, which shows again a bias towards Gram-positives in the approach. 

4. They found that many archaeal hydrolase genes are present on plasmids. Can they provide evidence that the hosts of these plasmids are archaea and not bacteria living in the same environment?

5. They should more clearly discuss that not all secreted peptidoglycan hydrolases might have a role in killing bacteria. Archaea might secrete hydrolases to digest the cell wall of lysed bacteria present in the environment, to access the nutrients. Did they check whether archaea that contain peptidoglycan hydrolases have genes encoding peptidoglycan uptake and processing factors?

6. They identified two peptidoglycan hydrolases in Halogramum solaris that seem to kill a halophilic bacterium. Can they add a scheme to indicate the genomic context of these hydrolase genes? What other genes are in their neighbourhood and are they part of an operon? 

Minor points:

1. Fig. 3G: should the axis label be colony forming units per ml? Can they explain the error bars and number of repeats in the figure legend?

2. Fig. 3H: the size bars are barely visible in the images and not explained in the figure legend. 

Reviewer #4:

This manuscript by Strock et al., which I'm assuming I received after the first round of revisions, presents a deep exploration of peptidoglycan hydrolases in Archaea as potential mediators of antagonistic interactions. They undertook a well thought-out integrative strategy, from phylogenetic analyses all the way to mass spectrometry and classic microbiological techniques when determining the activity of the J2ZZK6 peptidase. I wish to especially commend the authors on their stylistic approach and their essayist structuring and language, which I feel we ought to see more in manuscripts. Save for a few minor clarifications, I would strongly recommend that the manuscript be accepted almost as is.

1) Results

i) "phylogenetically balanced database": Where any genome quality criteria applied in the choice of genomes? Did the process of assembling the set of genomes result in certain environments being undersampled? Moreso for Bacteria; for Archaea the only undersampled taxon case I might expect are some genera in Methanobacteriales (maybe Methanosarcina or certain Halos?), though I would need to check the numbers for GTDB r214.

ii) "superphyla": There are no superphyla in the GTDB classification as of r220. Previously dubbed superphyla have been downgraded to phylum level (e.g., TACK to Thermoproteota, all Asgards to Asgardarchaeota). Iirc, only DPANN could be informally deemed a broadly accepted superphylum (and maybe Methanomada: Methanobacteriota+Methanobacteriota_A).

iii) "we find M23 C-terminal of a LysM domain": in the C-terminal direction/side of ... ?

2) Discussion

i) J2ZZK6: If the protein is zoocin-related, do any of the potential targets have a zoocin immunity factor or any identifiable putative defense systems? Another point is structure of J2ZZK6: the Alphafold prediction includes an intrinsically disordered N-terminal region. Most of that segment registers as NlpD on CDD but the sequence and disorder vaguely reminded of a Tat signal peptide. Tat signal peptides are found in halocins (https://doi.org/10.3389/fmicb.2012.00207) and afaik are common in Halobacteriales. A quick BLAST search shows no homologous segments in many Bacteria. I believe that a useful analysis for future work would be to classify the signal peptides of peptidoglycan hydrolases in Archaea and whether they were added, switched, or lost following the HGT event.

ii) "a valuable tool for identifying producer-target relationships": While I do agree with this statement, to what extent can the workflow be generalized to other peptidoglycan hydrolases? For example, do the corresponding databases and tools already exist?

iii) "act as keystone species": When mentioning keystone species, the (Moissl-Eichinger et al., 2018) review references (Tajima et al., 2001) and (Tapio et al., 2017), both of which discuss methanogens. Halos might be keystone species in their respective environments although I'm not readily cognizant of any literature on the subject. Generalizing to all Archaea without an analysis of the ecological distributions of PGHs seems like overreaching.

iv) "are frontier environments": This is fairly reasonable speculation but I would suggest a couple of additional bioinformatic analyses that could support it pre-emptively. First, when looking for potential PGH targets, for each GTDB species cluster, find the archaeal MAGs. Go to the corresponding SRA/BioProject and constrain the search to only bacterial MAGs therein. Other species in the cluster might also originate from frontier environments. Second, I would hypothesize that the halophily amino acid composition bias (D+E/I+K in https://doi.org/10.1038/s41564-024-01647-4) and/or the isoelectric point (https://doi.org/10.1016/j.mib.2015.05.009) of PGHs deviates from other secreted proteins, if they are active in frontier environments. 

3) Methods

i) "all named phyla were kept": Why? There is a wealth of genomes in certain "alphanumeric" phyla, for exampled SpSt-1190 and B1Sed10-29 in the DPANN.

ii) "less than 35% aligned residues": What was the rationale for the choice of percentage? Something in the literature or otherwise? I'm personally a fan of the smart-gap trimming mode in ClipKIT.

iii) "bacterium's NCBI taxonomy ID": Did you encounter any issues, due to GTDB-NCBI taxonomic discrepancies?

iv) "target proteins using ColabFold": How many proteins were in that set? Was ColabFold run locally?

v) "primary sequence or structural homology": In general, "homology" is not quantitative, so you would need to rephrase to "sequence identity" (did you mean "primary structure" or "primary and tertiary structure"?) and some equivalent for structures e.g., RMSD. 

Due to the broad interest of the topic for ecologists, evolutionary, and structural biologists, I hope that the authors will continue to further refine their approach. I would be interested in seeing more archaeal peptidoglycan hydrolases characterized and their structures solved, from environments where Archaea abound; from the gut microbiome to environments under severe energetic constraints, such as the deep subsurface.

---

## [Editor Report · Decision Letter 2]

Dear Toby,

Thank you for your patience while we considered your revised manuscript "Archaea produce peptidoglycan hydrolases that kill bacteria" for publication as a Research Article at PLOS Biology. This revised version of your manuscript has been evaluated by the PLOS Biology editors and the Academic Editor.

Based on our Academic Editor's assessment of your revision, we are likely to accept this manuscript for publication, provided you satisfactorily address the following data and other policy-related requests.

IMPORTANT - please attend to the following:

a) Many thanks for provinding the underlying data and code in Github (https://github.com/srom/archaea-vs-bacteria). However, because Github depositions can be readily changed or deleted, please make a permanent DOI’d copy (e.g. in Zenodo) and provide this URL in the Data Availabiity Statement and Figure legends (see below).

b) Please cite the location of the data clearly in all relevant main and supplementary Figure legends, e.g. “The data and code needed to generate this Figure can be found in https://zenodo.org/records/XXXXXXXX

c) Please make any custom code available, either as a supplementary file or as part of your data deposition (I suspect this is probably already in the Github/Zenodo deposition, but just checking!).

We expect to receive your revised manuscript within two weeks. 

*Published Peer Review History*

*Press*

Sincerely,

Roli

Roland Roberts, PhD

Senior Editor

rroberts@plos.org

PLOS Biology

CODE POLICY

We require the original, uncropped and minimally adjusted images supporting all blot and gel results reported in an article's figures or Supporting Information files. We will require these files before a manuscript can be accepted so please prepare and upload them now. Please carefully read our guidelines for how to prepare and upload this data: https://journals.plos.org/plosbiology/s/figures#loc-blot-and-gel-reporting-requirements

DATA NOT SHOWN?

---

## [Editor Report · Decision Letter 3]

Dear Toby,

Thank you for the submission of your revised Research Article "Archaea produce peptidoglycan hydrolases that kill bacteria" for publication in PLOS Biology. On behalf of my colleagues and the Academic Editor, Emily Eloe-Fadrosh, I'm pleased to say that we can in principle accept your manuscript for publication, provided you address any remaining formatting and reporting issues. These will be detailed in an email you should receive within 2-3 business days from our colleagues in the journal operations team; no action is required from you until then. Please note that we will not be able to formally accept your manuscript and schedule it for publication until you have completed any requested changes.

Sincerely, 

Roli

Senior Editor

PLOS Biology

rroberts@plos.org